# Application of Green Nanoemulsion for Elimination of Rifampicin from a Bulk Aqueous Solution

**DOI:** 10.3390/ijerph18115835

**Published:** 2021-05-28

**Authors:** Afzal Hussain, Wael A. Mahdi, Sultan Alshehri, Sarah I. Bukhari, Mohammad A. Almaniea

**Affiliations:** Department of Pharmaceutics, College of Pharmacy, King Saud University, P.O. Box 2457, Riyadh 11451, Saudi Arabia; wmahdi@ksu.edu.sa (W.A.M.); salshehri1@ksu.edu.sa (S.A.); sbukhari@ksu.edu.sa (S.I.B.); m.almaniea@gmail.com (M.A.A.)

**Keywords:** green nanoemulsion, rifampicin, in vitro characterizations, % removal efficiency, critical factors for adsorption

## Abstract

The study aimed to prepare green nanoemulsion (GNE) multi-components ((water/dimethyl sulfoxide–transcutol/isopropyl alcohol/capmul MCM C8 (CMC8)) to remove rifampicin (RIF) from a contaminated aqueous bulk solution. Pseudo ternary phase diagrams dictated several batches of GNE prepared following the reported method. Selected nanoemulsions (NF1–NF5) were characterized for morphology, globular size, size distribution (polydispersity index, PDI), viscosity, zeta potential, refractive index (RI), and free-thaw kinetic stability. They were investigated for percent removal efficiency (%RE) of RIF from the bulk aqueous solution for varied time intervals (10–60 min). Finally, scanning electron microscopy–energy dispersive x-ray (SEM–EDX) and inductive coupled plasma–optical emission system (ICP–OE) were used to confirm the extraction of trace content of dimethyl sulfoxide (DMSO) and others in the treated water. Considering the data obtained for globule size, PDI, viscosity, zeta potential, freeze–thaw stability, and refractive index, NF5 was the most suitable for RIF removal. The largest %RE value (91.7%) was related to NF5, which may be prudent to correlate with the lowest value (~39 nm) of size (maximum surface area available for contact adsorption), PDI (0.112), and viscosity (82 cP). Moreover, %RE was profoundly influenced by the content of CMC8 and the aqueous phase. These two phases had immense impact on the viscosity, size, and RI. The percent content of water, S_mix_, and CMC8 were 15% *w*/*w*), 60% *w*/*w*, and 25% *w*/*w*, respectively in NF5. SEM–EDX and ICP–OE confirmed the absence of DMSO and other hydrophilic components in the treated water. Thus, efficient NF5 could be a promising option to the conventional method to decontaminate the polluted aqueous system.

## 1. Introduction

Several reports have been published regarding drug(s) or pharmaceutical products, including chemicals, dyes, and cosmetics, affecting flora and fauna of aquatic ecosystems [1]. The presence of these products in (trace) surface drinking water and industrial effluents has already been well described (in published literature) in the last decade [2,3,4,5,6,7]. These contaminants (residual drugs in cosmetic sewage and in natural water) may be present—alone or as a mixture of drugs/pharmaceutical products—in waste water and surface water, resulting in adverse effects in aquatic life and humans. These residual drugs or contaminants remained unaltered in nature over time. Notably, antibiotics (among the residual substances) are potential contaminants because their metabolites have led to long-term systemic toxicity in aquatic ecosystems and human lives. Moreover, constant exposure challenged drug efficacy by developing resistance [8,9,10]. The magnitude of these contaminants are in ng/L or µg/L, depending on the source. Few drug or pharmaceutical products are anticipated to cause serious threats (diseases) to human health, but they may be causative factors in harming the environment. There are limited ecotoxicological data available for several pharmaceuticals, possibly causing serious adverse effects to humans and the ecosystem [1]. Notably, several pharmaceutical drug classes (anti-tubercular, antibiotics, anti-inflammatory, hormones, anti-diabetes, anti-cancers, antipsychotic antiseptic, topical products, lipid regulators, histamines blockers, and diagnostic media) have reported that these drug classes have been regularly monitored and assessed for their presence in waste water and surface water. Even these drugs and pharmaceutical products were detected in drinking water, as reported previously [11,12,13,14]. The presence of deleterious residual substances, particularly antibiotics (even at low doses) in aquatic systems, can generate several resistant strain types [15,16,17].

Two commonly used conventional methods (activated sludge and membrane bioreactor) were adopted for the treatment of the waste water to remove such drugs, pharmaceutical, and personal care products [5,6,10]. However, these methods were inefficient at removing contaminants up to a convincing level from the contaminated water (sewage, ground water, surface water, drinking water, and municipal water). Two recently reported methods to remove rifampicin from contaminated water are (a) heterogeneous (electrochemical) and (b) the Fenton reaction method (homogeneous method). The working principle of these methods are based on the generation of free radicals (hydroxyl group) using a powerful oxidizing agent [18]. Several new approaches have been reported to remove pharmaceutical contaminants from wastewater, such as a triboelectric generator, microalgae, nanofiltration, membrane bioreactor, polymeric coating, and piezoelectric electrospun nanofiber membrane [19,20,21,22,23].

Rifampicin (RIF) is the most clinically effective and established first-line anti-tubercular drug. The drug (molecular weight = 822.94 g·mol^−1^) is highly crystalline and hydrophobic (log *P* = 3.8) with limited aqueous solubility (~1–2 mg/mL soluble in water) [24,25]. The drug is associated with peripheral neuropathic toxicity, chemical instability in the stomach (~32% as desacetylrifampicin), and limited oral absorption [26,27,28]. After oral or parenteral administration, RIF is biotransformed into several deacetylated metabolites in the body and excess metabolites are eliminated through urine. It was reported in the literature that the drug was detected in effluents treated from wastewater treatment plants (WWTPs) and the treatment system failed to eliminate this type of compound [29,30]. Methicillin resistant *S. aureus* (MRSA) isolates revealed resistance to broad spectrum potential antibiotics. As compared to MRSA isolates obtained from WWTPs, S1/S2 isolates showed a broad resistance pattern, particularly against gentamicin (30.4%) and, to a lower extent (6.3%) against ceftaroline, tigecycline, and rifampicin [31]. Recently, rifampicin degradation was investigated by using “Advanced Oxidative Processes (AOPs)” [32,33]. This is a matter of concern—that the potential ecotoxicity and residual occurrence in the environment challenged recent methods and the quality of human life.

Green nanoemulsion multi-components are isotropic and transparent mixtures of water, lipid, surfactant, and co-surfactants. These systems are stable, low viscous, cost effective, and scalable for large-scale treatment of wastewater, maximum drug solubilization, and drug extraction capability [34]. The benefits of such lipid-based adsorbents have rarely been investigated to eliminate toxic dyes (mostly from textiles), metals, residual, and organic pharmaceutical products from wastewater and bulk aqueous solutions. In the literature, green nanoemulsion (GNE) was not investigated as an effective approach to eliminate anti-tubercular rifampicin, toxic contaminants, and cosmetic products from wastewater or bulk aqueous solutions. We aimed to prepare water/DMSOT/2-propanol/capmul MCM C8 (CMC8) GNE to remove anti-tubercular RIF from a bulk water solution for the first time. Presently, the selected excipients were of the green/bio-safe category. These were water (aqueous phase), DMSOT (dimethyl sulfoxide containing 10%v/v transcutol as surfactant), IPA (isopropyl alcohol as co-surfactant), and CMC8 (oil phase) to remove RIF through the liquid–liquid adsorption mechanism by GNE.

## 2. Materials and Methods

### 2.1. Materials

The drug was a macrocyclic antibiotic (Figure 1) received (as a generous gift sample) from Unicure India. Isopropyl alcohol (IPA), dimethyl sulfoxide (DMSO), methanol, and acetonitrile were procured from Merck, (Mumbai, India). Capmul MCM C8 (CMC8) was received as a gift sample from BASF. Transcutol HP is, chemically, diethylene glycol monoethyl ether obtained from Gattefossé (Lyon, France).

### 2.2. Methods

#### 2.2.1. Analysis

Rifampicin was assayed using a previously established method [25]. The drug content was estimated using a RP-C_18_ column (150 × 4.6 mm, 5 μm, Purospher Star, Merck, Germany). A UPLC analysis system was probed with a rotary-vacuum pump (E2M30, West Sussex, UK) and a N_2_ generator (Killearn, UK) to elute RIF. In the analysis of the drug, the mobile phase was composed of acetonitrile (80% *v*/*v*) and ammonium acetate buffer (2 mM) (20% *v*/*v*). In order to avoid bubbles and insoluble fibers, the mobile phase was filtered using a membrane filter and subsequently bath sonicated. The samples were analyzed over 5 min of run time (1.0 mL min^−1^) and injection volume of 20 µL. A working linear calibration curve was established (12.27–7999.92 ng/mL) with a regression coefficient (r^2^) of 0.9998. The analysis was replicated to obtain mean and standard deviations.

#### 2.2.2. Pseudo-Ternary Phase Diagrams (PTDs) and Nanoemulsions

Several GNEs (water in oil) were prepared using water, DMSOT (DMSO containing constant amount of transcutol, 10% *v*/*v*), isopropyl alcohol (IPA), and CMC8, as per the reported method [20,35]. Several batches of green nanoemulsions were prepared using the oil phase titration method [35]. Notably, various trial nanoemulsions were developed using neat transcutol, IPA, and capmul. However, these were not poorly stable. Therefore, DMSO containing 10% transcutol (DMSOT) and IPA worked as a suitable surfactant and cosurfactant in preparing nanoemulsions, respectively. Both were properly blended to obtain a S_mix_ ratio (2:1, 1:2, 1:1, and 1:3). To construct several PTDs, the aqueous phase and the S_mix_ ratio were mixed completely in varied mass ratios (from 9:1 to 1:9). Now, the obtained mixture was slowly titrated with CMC8 (dropwise). Each product was carefully inspected for stable and isotropic nanoemulsion. Few selected nanoemulsions and their compositions are revealed in Table 1. Nanoemulsion with any signs of instability (cracking, phase separation, creaming, and inconsistent) were omitted for further studies.

#### 2.2.3. Evaluation Parameters of Nanoemulsions

Prepared GNEs were evaluated for their stability (kinetically stable system), globular size (radius), size distribution (PDI), zeta potential, RI, and viscosity (η). NF5 was separately dispersed in plain water and drug solution for 30 min, which were inverted into o/w type of nanoemulsion. Both were evaluated for morphological assessment of the nanoemulsion globule using high-resolution transmission electron microscopy (HR-TEM) (H-7500, Hitachi, Japan) at an ambient temperature and 200 k voltage. For this, the sample (a drop) was negatively stained with phosphotungstic acid (0.1% *w*/*v*) and dried overnight by placing on the copper grid. Then, the excess sample was removed using an adsorbent before coating to make it conductive. The images were scanned under high-resolution. The stability study included a series of cyclic processes (freeze–thaw cycles) (heating, cooling, and centrifugation) following the previous method [34,35,36]. Particle size (radius) and PDI values were assessed by a Malvern instrument (Zetasizer Nano ZS, Worcestershire, UK). The samples were previously diluted (100 folds) with water to avoid errors in analysis. Rheological behavior (viscosity) was determined using the Bohlin viscometer (Bohlin Visco 88, Malvern, UK). The viscometer was assembled with a coaxially arranged cone and plate. The sample was kept on the plate. The cone was lowered down to the plate and allowed to move for a fixed period of time. The study was repeated for mean values. The RI values of freshly prepared nanoemulsion and pure lipid (CMC8) were measured using a refractometer (Abbe type) (Bausch and Lomb, Optical Company, Rochester, NY, USA).

#### 2.2.4. Prepared Stock Solution: For a Calibration Curve

RIF is poorly soluble in water (~1–2 mg/mL) at 25 °C, whereas it is soluble in CMC8, DMSOT, and isopropyl alcohol [24,25]. Therefore, we added DMSOT (5% *v*/*v*) in an aqueous medium for stock solution. An accurate amount of the drug was solubilized in a hydroalcoholic solution to prepare a stock solution of RIF (100 ppm). A range of concentration (5.0–40.0 ppm) was prepared using the stock solution through serial dilution. RIF concentration was estimated using an UPLC at an absorbance wavelength of 337 nm [25].

#### 2.2.5. An Adsorptive Study: RIF Removal from Aqueous Solution

Freshly prepared nanoemulsion was accurately weighed (weight “m” = 1 g), a minor quantity in comparison to the aqueous RIF solution (10 mL containing 100 ppm). Now, the weighed nanoemulsion was completely dispersed in 10 mL (V) of the aqueous system to separate RIF from the stock solution (100 ppm/10 mL). Furthermore, the mixed nanoemulsion–drug mixture was forcibly vortexed for 15 min and left for a given time period at 25 °C (benchtop standing). Thus, the drug solution was exposed with green nanoemulsion for different time points (10–60 min). The mixed system was destabilized by freezing (−21 °C for 3 h) followed by heating at 60 °C for 2 h; this resulted in two separated phases [37]. Finally, this was subjected to high centrifugation (10,000 × *g* for 10 min) to obtain clear water as a supernatant and oily phase at the bottom of the tube. The separated oil phase was employed to determine RIF content by diluting in methanol. Hence, the content of the drug (Q in ppm/g at time t), adsorbed to the oil globules of nanoemulsion (drug–liquid adsorption) at specific time points (10, 30, and 60 min), estimated using Equation (1):Q = [(C − C_t_)/m] × V(1)

Similarly, Equation (2) was used to calculate %RE: %RE (removal efficiency) = [(C − C_t_)/C] × 100(2)

In the equations, C and Ct represent the added content of the drug and the estimated content of RIF after specified time (t), respectively [38].

#### 2.2.6. Assessment of Treated Water

To find the probable chance of extracted DMSO, transcutol, and other water soluble components of NF5 in the aqueous phase (treated water), the treated water was evaluated using SEM–EDX (scanning electron microscopy–energy dispersive X-ray) ((JEOL JSM-6390LV, Tokyo, Japan) and ICP–OE (inductively coupled plasma-optical emission spectrometry) (Thermo Fisher Scientific, Bremen, Germany). UV spectroscopy and IR spectroscopy were insufficient due to detection limit. Therefore, these two advanced technologies were applied to find elemental analysis of the compounds (DMSO, transcutol, and others). ICP–OE provided elemental analysis, which was supportive of the SEM–EDX findings.

## 3. Results and Discussion

### 3.1. Preparation of GNE by PTDs

Several GNEs were prepared as dictated by the constructed PTDs, illustrated in Figure 2. It is clear from the PTDs that there was an impact of the surfactant to co-surfactant ratio to delineate maximum nanoemulsion regions (water/DMSOT/IPA/CMC8 nanoemulsion) (Figure 2). It is also clear from Figure 2 that maximum solubilization of the aqueous phase (67.4% *w*/*w*) by S_mix_ ratio was achieved at the S_mix_ ratio of 1:3 (surfactant to co-surfactant ratio, 19.7% *w*/*w*) (Figure 2A). This solubilization decreased with a decreased concentration of co-surfactant IPA over DMSOT (surfactant) (1:1 < 1:2 < 1:3), as shown in Figure 2B,C. Notably, the nanoemulsion zone slightly decreased at the S_mix_ ratio of 2:1, as compared to the DMSOT: IPA ratio of 1:1 (Figure 2D). Thus, the highest solubilized content of the aqueous phase is 50.5% *w*/*w* at the S_mix_ of 37.8% *w*/*w* (2: 1 ratio). The highest value of zone delineated (covered) for the nanoemulsions was obtained with the “1:3” ratio (S_mix_). Therefore, the various compositions of GNEs (NF1 to NF5) were screened from the S_mix_ ratio of 1: 3 (Figure 2A) at a fixed content of S_mix_ (80% *w*/*w*), and varied amounts of water phases (2 to 16% *w*/*w*), as shown in Table 1.

### 3.2. Characterizations of Water/DMSOT/IPA/CMC8 GNEs

The morphological assessments of dispersed NF5 were carried out using HR-TEM with plain water and the drug solution after 30 min (without destabilization). The HR-TEM images are illustrated in Figure 3A,B, where there is no globular aggregation after dispersion in both mediums. It is apparent that the developed nanoemulsion NF5 was efficient to invert from the w/o to o/w type of nanoemulsion after dispersion. GNEs are well established carrier systems and considered kinetically stable systems. Therefore, we performed this study at various temperatures (series of cycles) and subsequent centrifugation. This study helped screen out metastable and unstable nanoemulsions prepared from established compositions. Fortunately, the explored composition and excipients were substantial at developing kinetically stable green nanoemulsions, such as NF1 to NF5. These were carefully inspected after exposure to stress conditions of temperature and centrifugation steps. All of them were found stable and clear, as shown in Table 2. Thus, the explored ratio of S_mix_ might be capable of developing a firm protective layer around the nanoglobules to protect nanoemulsion from coalescence. All of the developed nanoemulsions were found stable with zeta potential from −24.9 to −29.3 mV. There was a slight reduction in zeta potential from NF1 to NF5, which may be due to a decreased concentration of CMC8 (capric and caprylic acid bearing carboxylic functional group at terminal of hydrocarbon chain).

Fundamentally, a surface area is directly proportional to the size of the particle. This concept can easily be applied to removal efficiency of the drug from an aqueous solution using nanoemulsion. The particle size and size distribution had quite a desirable impact on %RE. The smaller the size, the larger the contact of RIF to oil globules of nanoemulsion after exposure. Thus, the globular size of nanoemulsion is a critical and prime factor controlling the stability and %RE of RIF from an aqueous solution. Thus, the adsorption of RIF to the largely exposed surface area of nanoemulsion results in increased %RE due to maximum drug–oil solubilization through lipophilic–lipophilic interaction. The organic oil phase and aqueous continuous phase had impact on the globular size (38.78–98.65 nm) and PDI as shown in Table 3. The size was found to be increased with an increase in the CMC8 content, whereas it was decreased when the content of the aqueous portion was increased. This is obvious due to a relatively high concentration of S_mix_ in NF5 as compared to CMC8, resulting in a reduced globular size at a high content of the aqueous phase and low content of CMC8 in water/DMSOT/IPA/CMC8 nanoemulsion (Table 3). These findings comply with the previous literature, where lipophilic indomethacin was eliminated using the green nanoemulsion approach (water/transcutol/IPA/Capryol) from the bulk aqueous system [34,35,36].

NF5 and NF1 exhibited the lowest (~39 nm) and the largest (~99 nm) values of globular sizes, respectively. A similar pattern was obtained for PDI values (0.112 in NF5 and 0.283 in NF1) (Table 3). The NF5 exhibited better size distribution as evidenced with PDI value compared to NF1, which may be due to relative content of CMC8 in nanoemulsion. The PDI values of NF1–NF5 were in the range of 0.112–0.283. Thus, NF5 elicited the lowest value of size due to a minimum concentration of CMC8 and a maximum aqueous phase as compared to NF1. A graphical illustration was portrayed, suggesting an effect of aqueous and CMC8 content on the globular size of NEs (Figure 4A). The result of viscosity ranged as 81.9–138.9 cP (Table 3) for all nanoemulsions (NF1–NF5). Generally, oil or lipid is comparatively more viscous than aqueous system or emulsion. Therefore, the values of viscosity increased with an increase in CMC8 content in nanoemulsion, and vice-versa. This was also related to the content of the aqueous phase and globular size. A high content of water (relative to CMC8) and reduced globular size caused lower value of viscosity. Thus, NF1 had the maximum value of viscosity (~139 cP) whereas NF5 had the lowest value of viscosity (~82.0 cP). Figure 4B illustrated the impact of oil (CMC8) and aqueous phases on the viscosity behavior. A similar pattern of results was obtained when caffeine, 5-fluorouracil, and indomethacin were removed from the bulk aqueous solution using green nanoemulsion [34,39,40]. The RI is an optical property of a nanoemulsion. In general, nanoemulsion possessing RI value close to 1.33 (refractive index of water) is considered transparent, stable, and isotropic. The RI value of CMC8 was 1.43, which is relatively higher than water. However, the emulsified CMC8 in nanoemulsion results in a significant reduction in RI value (1.43), suggesting a small globular size in nanoemulsion. The values obtained from NF1–NF5 ranged from 1.325 to 1.381, which is quite close to 1.33 (Table 3). Water in oil type of nanoemulsions showed RI values close to 1.4 due to CMC8 as a continuous phase.

### 3.3. Removal Efficiency and Impact of Factors Affecting Adsorption of RIF

RIF is an established drug with known poor aqueous solubility and pharmaceutical contaminants in the aqueous system. The drug is slowly accumulated in flora and fauna of aquatic ecosystems, to a toxic level, over a long period of time. Finally, this causes serious adverse effects on human beings after consuming treated water and aquatic foods. The removal of RIF depends on the physicochemical properties of RIF, the nature of the lipid of nanoemulsion (lipophilicity and interaction), contact time, and specific features of green nanoemulsion (globule size, surface area, and viscosity). A larger surface area is obtained with reduced sizes of any materials or particles. Complete removal of pharmaceutical drugs from aqueous systems is challenging and critical for conventional methods [41]. Moreover, there are few factors, such as solubility, hydrophobicity, biodegradability, adsorption ability, and volatility of the drug, which have major impact on %RE [42,43,44]. Figure 5 shows the impact of CMC8 and water content on %RE of NF1–NF5. Notably, the %RE of RIF increased when the content of water increased (3–15% *w*/*w*), whereas it rapidly decreased when increasing the content of CMC8 (25–37% *w*/*w*), as shown in Figure 5.

Thus, removal of RIF from a contaminated aqueous system is an extremely tedious task due to poor aqueous solubility and weak adsorption onto the conventional adsorbent. Therefore, we assessed %RE of green nanoemulsion (water/DMSOT/IPA/CMC8 nanoemulsions) (NF1–NF5) to eliminate RIF from a bulk aqueous system. Removal efficiencies of NF1–NF5 were estimated at 10, 30, and 60 min (different contact times). In contrast, NF1 showed the lowest value of %RE after 10 min of exposure time as compared to NF2–NF5. The results of adsorption efficiency (%RE) of RIF onto the surface of green nanoemulsions “NF1–NF5” are presented in Table 4. The estimated values of %RE of NF1–NF5 were found in the range of 75.1–95.7% at 60 min (Table 4). NF5 was found to have the maximum %RE after 60 min of exposure as compared to NF1–NF4. There was a significant impact of CMC8 and water content of nanoemulsions (NF1–NF5) on the %RE of RIF. This finding may be correlated with the characterization parameters of nanoemulsions, such as globular size and viscosity. There were significant impacts of viscosity and globule size on %RE, whereas the exposure time (contact time) had a negligible impact, as shown in Figure 6A–C. Nanoemulsion is an isotropic system with maximum stability, capable of emulsifying promptly in the aqueous system. Therefore, nanoemulsion with a small globule size offers a greater surface area for adsorption of RIF onto the nanoglobules as the adsorptive surface and vice-versa. Moreover, nanoemulsions of low viscosity may result in rapid emulsification and exposure of adsorptive surface available for adsorption. Figure 6A exhibited the impact of viscosity on %RE of RIF from an aqueous system. Thus, the %RE value was observed to be decreased rapidly when the values of viscosity (Figure 6A) and the globule size (Figure 6B) were increased. Thus, NF5 revealed the highest %RE (91.7%), which may likely be due to the lowest value of viscosity (81.5 cP) and the smallest size of globules (38.9 nm). This result was further supported by establishing a relationship between the square of the globular size and %RE of NF1–NF5. Figure 7 illustrates that %RE constantly decreased with the increase in the globular size. Hence, the results obtained are in accordance with previous reports, suggesting the removal of lipophilic contaminants/drug/cosmetics/dyes [34,36,41,42,43,44].

### 3.4. Proposed Mechanistic Perspective

As we discussed earlier, removal efficiency of RIF from an aqueous solution depends on several factors. The mechanism of removal of RIF using nanoemulsions (NF1–NF5) is the result of them working together (insolubility, hydrophobicity, relative diffusion, and surficial adsorption). In addition, the adsorption of RIF was dependent on the content of CMC8 and water. It is well known that CMC8 is a hydrophobic medium chain triglyceride. On the other hand, RIF is extremely insoluble in water. Therefore, on exposing nanoemulsion with the aqueous bulk solution, it must be loaded in the oil phase of green nanoemulsion due to lipophilic nature and relatively high solubility of RIF in CMC8. There is a probable chance of a relative diffusion of RIF between the bulk aqueous solution and the oil phase of green nanoemulsion when they come in contact with each other. Furthermore, freeze–thaw and heating to a high temperature resulted in destabilization of nanoemulsion in the form of two phase separations. RIF in the oil phase was quantified for RIF content. The aqueous phase was further assessed for presence of extracted DMSO and other components of nanoemulsion. Thus, the drug solubility in CMC8 and subsequent relative diffusion could be a possible mechanism for removing RIF from an aqueous solution. This is illustrated in Figure 8. A similar mechanistic view was proposed to remove lipophilic indomethacin using green nanoemulsion [34,35,36].

### 3.5. Assessment of Treated Water

Chemically, DMSO is dimethyl sulfoxide containing sulfur as an element. ICP–OE was used to detect the presence of elements in the treated water. It was expected that the presence of sulfur in treated water may confirm extraction of DMSO from nanoemulsion. C, H, and O are common elements present in water, transcutol, and DMSO. Therefore, these elements were kept out of detection due to interference in results. Similarly, the SEM–EDX method was used to detect elements, excluding the common elements. The results are presented in Figure 9. Figure 9A is the result of SEM–EDX, which exhibited several peaks of intensity of elements present in the treated water. The most intense peak was observed for Ca, Na, and K. This qualitative assessment confirmed that these alkali and alkali-like metals are due to hardness of water (dissolved salt). However, Fe, Ti, and Cu were extremely low. Cu may be due to the copper grid used during analysis. Ti and Fe were insignificant. Notably, the treated was free from any sulfur element, suggesting absence of DMSO extraction or the content might be below detection. These findings were further supported with quantitative assessment using ICP–OE.

The result of ICP–OE is portrayed in Figure 9B where Ca (0.089 ppm), K (0.004 ppm), Mg (0.042), and Na (0.048) were considerably high in the treated water. However, Fe, Ti, Zn, Mo, and Cu were below the detection limit (BDL). The absence of sulfur in Figure 9B confirmed the absence of DMSO extraction below the detection limit. The treated water was also scanned using UV–Vis spectrophotometer and IR spectroscopy to identify absorbance wavelength and finger printing of functional groups of DMSO, transcutol, and other components, respectively (data not included here). These approaches could not detected due to the content below BDL or the absence of these. Conclusively, the separated organic phase included CMC8 and S_mix_, leaving the water phase behind.

### 3.6. Future Perspectives

RIF is a commonly used anti-tubercular drug to control tuberculosis, inside and outside of hospitals. Pharmaceutical contaminants can be removed using the green nanoemulsion approach, composed of safe excipients at economic cost. The proposed approach can easily be scaled-up for large-scale industrial processes. However, there are several variables and evaluation parameters that need to be investigated. In this approach, we used DMSOT and volatile IPA, which needed to be removed (extracted) to clean the portion of treated water. Notably, the content of RIF that remained in treated water was below the detection level of the UPLC analysis method, suggesting an efficient removal process of the proposed approach. This approach can be used as a preliminary step before being subjected to advanced reported techniques to improve removal efficiency. Several authors reported on a new methodology to remove pharmaceutical contaminants from wastewater, such as (a) triboelectric generator, (b) microalgae, (c) nanofiltration membrane bioreactor, (d) polymeric coating, and (e) piezoelectric electrospun nanofiber membrane [19,20,21,22,23]. Khushwaha et al. developed a triboelectric generator as an efficient and green approach (80% degradation efficiency achieved using the triboelectric effect) to treat wastewater [19]. Microalgae was also reported for this purpose. A well-known green approach (over the last few decades) has been utilization of *Nannochloropsis spp.* (grown in fresh and sea water) for nutraceutical and food supplements [45]. Furthermore, this microalga has been used to remove pharmaceuticals (paracetamol, ibuprofen, olanzapine, and simvastatin) in free and immobilized form [20]. Both forms responded differently to each pharmaceutical. Zaviska et al. reported the use of a nanofiltration membrane bioreactor to achieve high purity water, processed from contaminated water, with ciprofloxacin and cyclophosphamide [21]. Thus, these are emerging and promising alternative approaches to conventional techniques, which may be further amplified if assembled with nanoemulsion-based pre-treated water.

## 4. Conclusions

Rifampicin is a potential anti-tubercular drug used to treat tuberculosis. Contaminated water with potential drugs challenge the flora and fauna of aquatic lives and human beings. Conventional approaches are unable to eliminate these potential micro-pollutants present in effluents. In this study, we investigated green nanoemulsion composed of biocompatible excipients to remove dissolved RIF from the aqueous solution. The proposed method was validated by assessing the content of RIF removed from a stock solution. There are several factors controlling removal efficiency of RIF from the aqueous system. However, the content of CMC8 and water had profound impacts on the %RE of RIF from the aqueous solution. Other factors (globular size, PDI, and viscosity) were also taken into account for %RE.

The present approach focused on removing RIF from contaminated water. The proposed mechanism of RIF removal was a preliminary step of our research design. In future publications, the retained amount of DMSOT and IPA needs to be removed, using a conventional approach sequentially assembled with this step. However, several new methodologies can be assembled sequentially with the green nanoemulsion approach to amplify removal efficiency, as discussed in “Future Perspectives”. A complete proof-of-concept was generated to ensure the suitability of nanoemulsion over conventional approaches. Conclusively, a nanoemulsion-based approach was simple, scalable, economic, and efficient as compared to traditional techniques.

## Figures and Tables

**Figure 1 ijerph-18-05835-f001:**
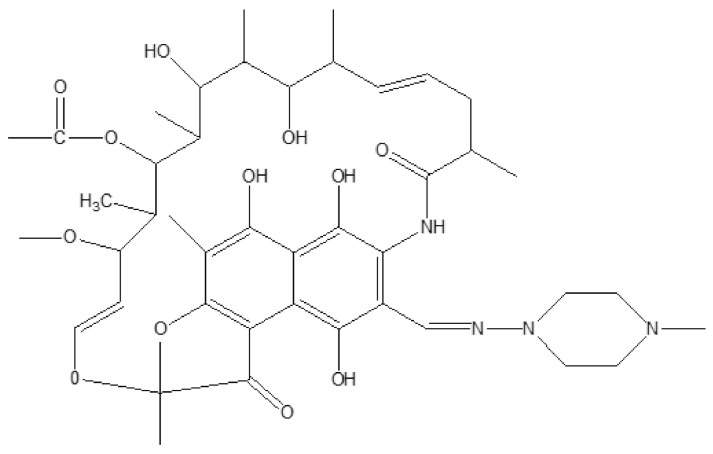
Chemical structure of rifampicin.

**Figure 2 ijerph-18-05835-f002:**
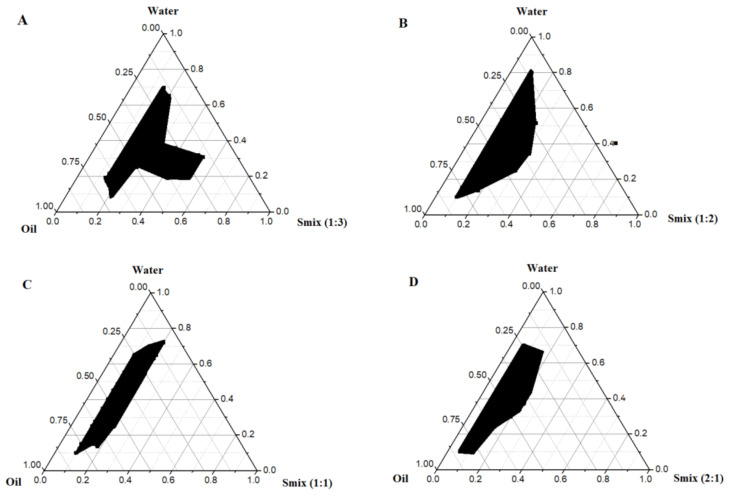
Pseudo ternary phase diagram of rifampicin loaded nanoemulsions with varied ratios of S_mix_: (**A**) S_mix_ ratio of 1:3, (**B**) S_mix_ ratio of 1:2, (**C**) S_mix_ ratio of 1:1, and (**D**) S_mix_ ratio of 2:1.

**Figure 3 ijerph-18-05835-f003:**
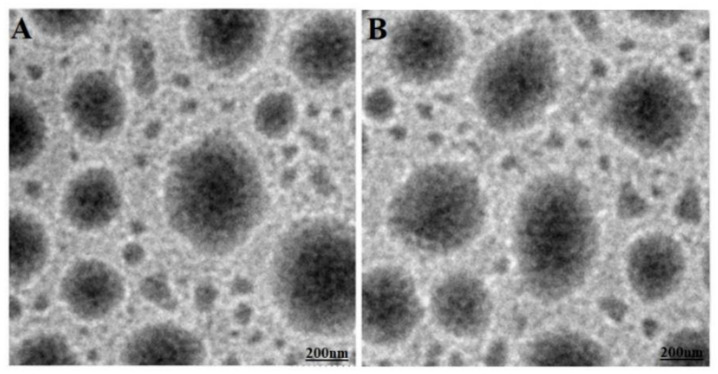
(**A**) Transmission electron microscopy (TEM) image of the dispersed o/w type of NF5 in water after 30 min, and (**B**) TEM of dispersed NF5 (o/w) in the drug solution after 30 min. Both exhibited globular sizes below 100 nm at 15,000× (magnification).

**Figure 4 ijerph-18-05835-f004:**
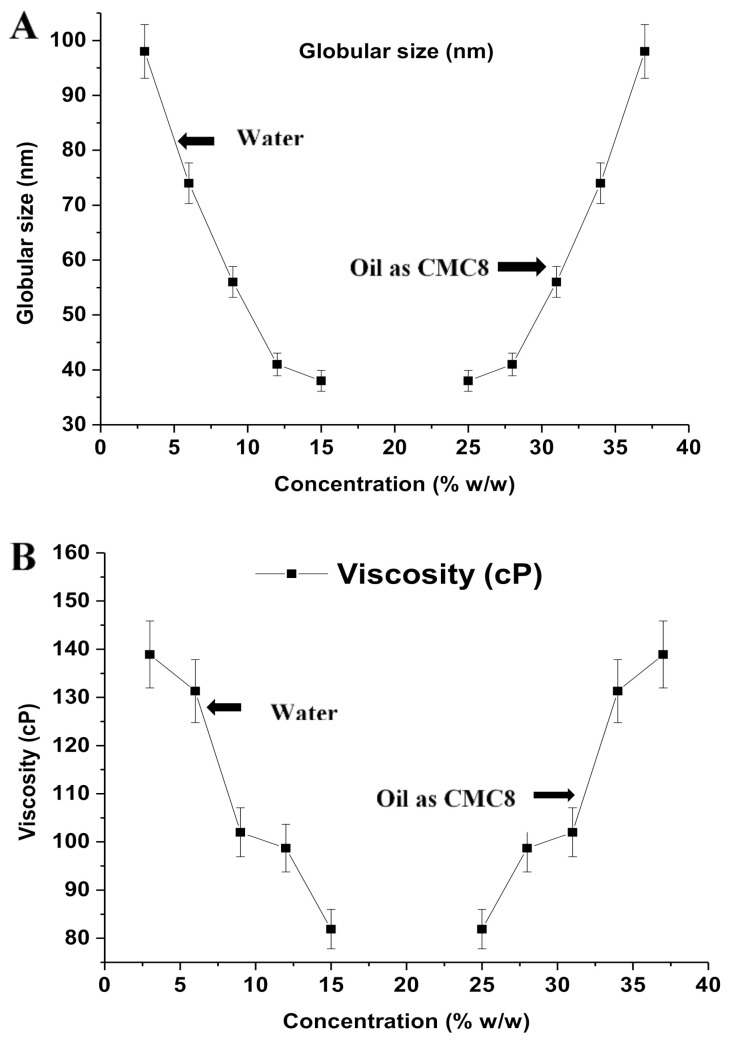
(**A**) The effect of continuous organic phase (CMC8) and internal aqueous phase (water) concentrations on the globular size of water/DMSOT/IPA/CMC8 nanoemulsions, and (**B**) the effect of continuous organic phase (CMC8) and internal aqueous phase (water) concentrations on viscosity (cP) of water/DMSOT/IPA/CMC8 nanoemulsions.

**Figure 5 ijerph-18-05835-f005:**
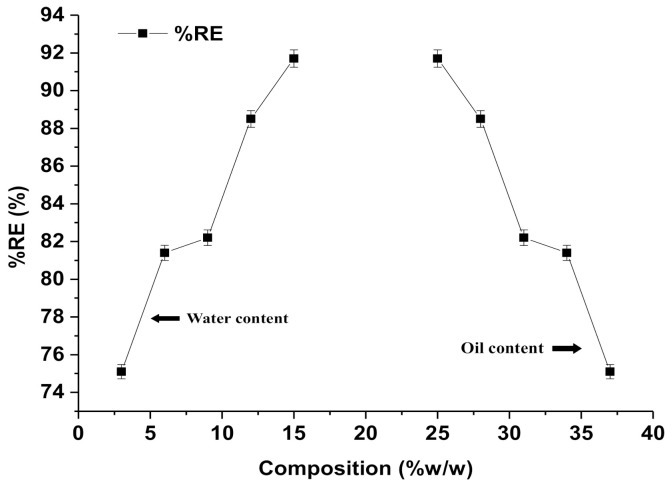
Effect of aqueous and oil phase concentration on %RE of RIF from the aqueous bulk solution through water/DMSOT/IPA/CMC8 nanoemulsions.

**Figure 6 ijerph-18-05835-f006:**
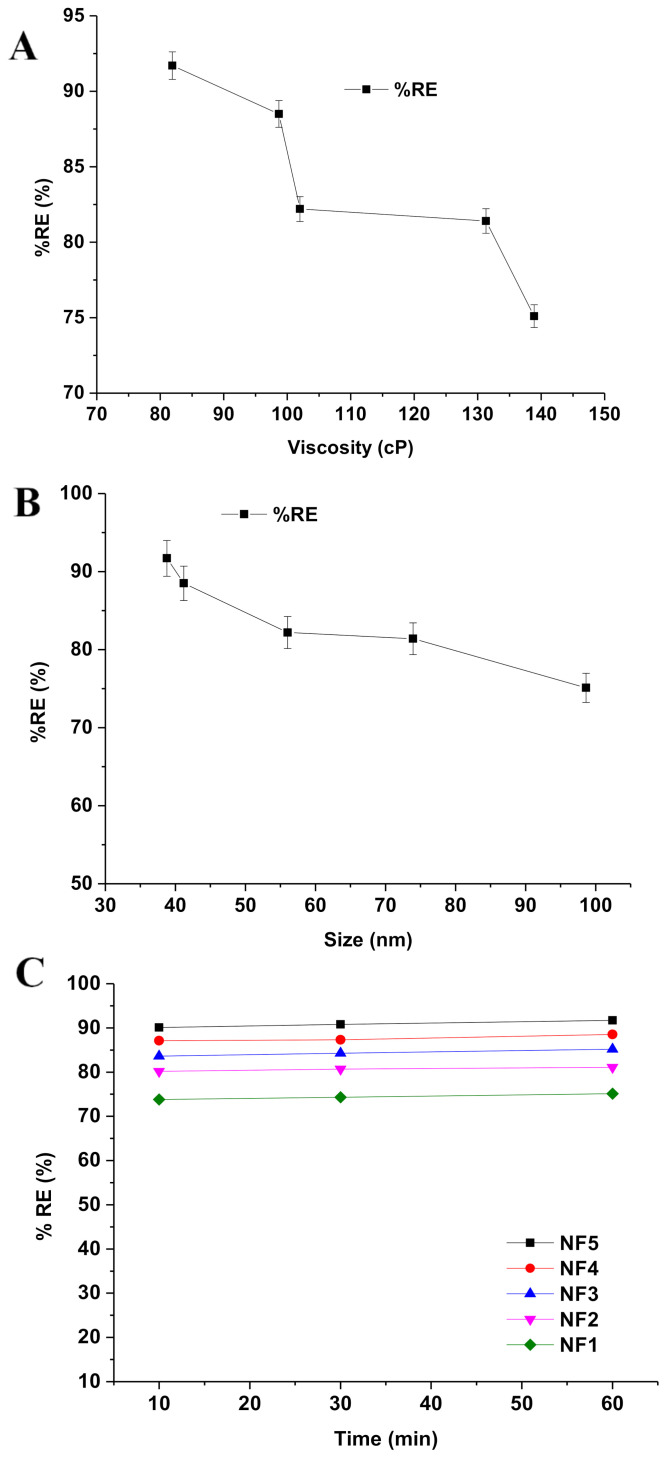
Factors affecting %RE of RIF from an aqueous solution: (**A**) impact of viscosity (cP); (**B**) impact of globule size (nm); and (**C**) impact of exposure time (min).

**Figure 7 ijerph-18-05835-f007:**
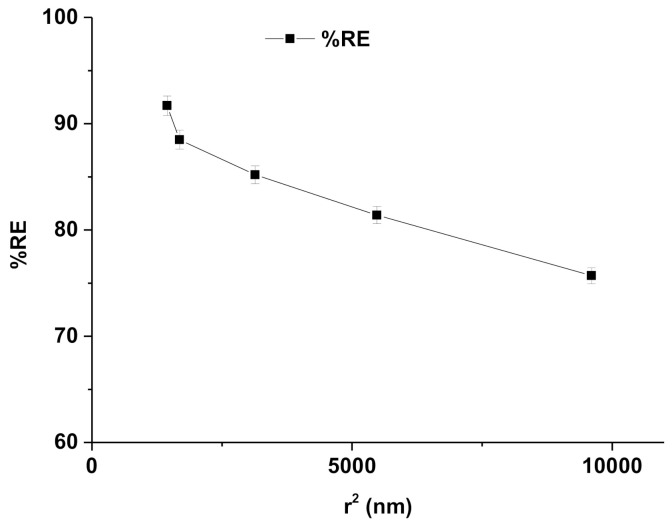
Effect of r^2^ on %RE of RIF from aqueous bulk solution through water/DMSOT/IPA/CMC8 nanoemulsions.

**Figure 8 ijerph-18-05835-f008:**
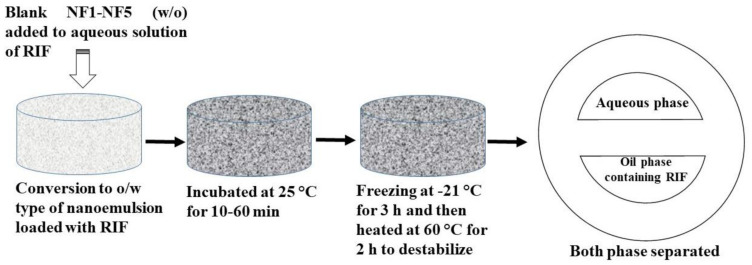
A schematic illustration of a proposed mechanistic perspective for removal of insoluble rifampicin from an aqueous bulk solution using a green nanoemulsion. After dropping of nanoemulsion to the aqueous drug solution under stirring, there is a probable chance of drug adsorption onto the oil (CMC8) globule surface during incubation at 25 °C. Freeze–thaw (−21 °C and subsequent heating at a high temperature (60 °C) resulted in two phase separation (destabilization). The oil phase, with adsorbed (loaded with RIF), was separated by centrifugation for quantitative analysis.

**Figure 9 ijerph-18-05835-f009:**
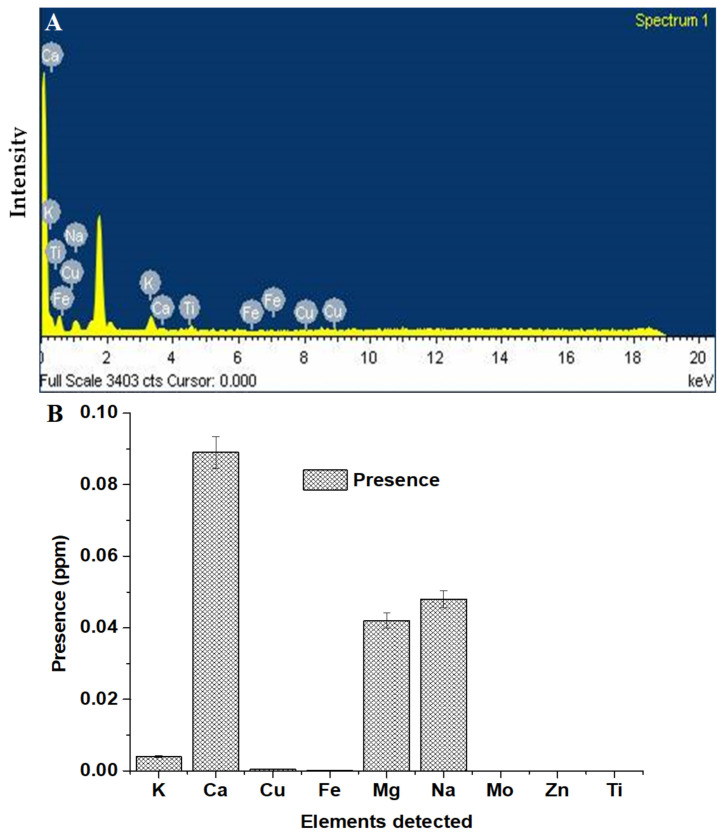
(**A**) SEM–EDX result of the treated water for qualitative assessment, and (**B**) ICP–OE result of the treated water for detection of sulfur (as an indirect assessment) as an element present in DMSO.

**Table 1 ijerph-18-05835-t001:** Summary of nanoemulsions (water/DMSOT/IPA/CMC8) composition.

Code	Composition (%*w*/*w*)				
	Water	DMSOT	IPA	CMC8	S_mix_
NF1	3	15	45	37	1:3
NF2	6	15	45	34	1:3
NF3	9	15	45	31	1:3
NF4	12	15	45	28	1:3
NF5	15	15	45	25	1:3

Note: IPA = isopropyl alcohol, CMC8 = Capmul MCM C8.

**Table 2 ijerph-18-05835-t002:** Kinetically stable nanoemulsions (water/DMSOT/IPA/CMC8).

Code		Stability Cycles			
	Zeta Potential (mV)	Centrifugation	Heating (40 °C)	Freezing (−21 °C)	S_mix_
NF1	−29.3	√	√	√	1:3
NF2	−27.8	√	√	√	1:3
NF3	−27.1	√	√	√	1:3
NF4	−25.4	√	√	√	1:3
NF5	−24.9	√	√	√	1:3

Note: √ mark indicates passed; there was no sign of physical instability.

**Table 3 ijerph-18-05835-t003:** Results of nanoemulsions (water/DMSOT/IPA/CMC8).

Code	Characterization Parameters (Mean ± Standard Deviation)			
	Globular Size (nm)	PDI	Viscosity (η) (cP)	Refractive Index
NF1	98.65 ± 11.5	0. 283	138.9 ± 10.4	1.381 ± 0.004
NF2	73.92 ± 8.4	0.252	131.3 ± 9.3	1.374 ± 0.003
NF3	56.03 ± 7.1	0.241	102.0 ± 8.6	1.355 ± 0.006
NF4	41.19 ± 4.9	0.189	98.7 ± 7.12	1.351 ± 0.008
NF5	38.78 ± 1.2	0.112	81.9 ± 5.6	1.325 ± 0.001

Note: PDI = polydispersity index, CMC8 = Capmul MCM C8.

**Table 4 ijerph-18-05835-t004:** Results of %RE on to the surface of nanoemulsions (water/DMSOT/IPA/CMC8).

Code	%RE at Varied Time of Exposure (min)		
	10	30	60
NF5	90.1 ± 0.454	90. 8 ± 0.451	91.7 ± 0.456
NF4	87.1 ± 0.439	87.3 ± 0.441	88.5 ± 0.443
NF3	83.6 ± 0.428	84.3 ± 0.436	85.2 ±0.431
NF2	80.2 ± 0.411	80.7 ± 0.409	81.4 ± 0.413
NF1	73.8 ± 0.383	74.3 ± 0.387	75.1 ± 0.384

Note: %RE = percent removal efficiency.

## Data Availability

The data were generated during the study.

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
