# Peer review of "Application of Green Nanoemulsion for Elimination of Rifampicin from a Bulk Aqueous Solution"

_ijerph, 2021, doi:10.3390/ijerph18115835_

Round 1
Reviewer 1 Report
Attached in the word file

Author Response
Response letter Date: 15-05-2021
Reviewer 1
Comment 1): Why DMSO was chosen as a surfactant? Is there any information in the literature about DMSO- stabilized nanoemulsions? Does it have proven and sufficient surface active properties? If yes, please cite appropriate literature.
Response 1: Thank you for pointing out missing content. In this study, DMSO contains a fixed concentration of transcutol as surfactant (10%v/v, and expressed as DMSOT). Therefore, experimental section was revised in the revised manuscript. Moreover, the title was revised to avoid this confusion. During exhaustive phase diagram study, we come to know, neat transcutol was capable to give limited zone of nanemulsion (data not included in this study). Faiyaz et al. reported successful formation of green nanoemulsion to remove diclofenac from aqueous solution (Shakeel, F., Haq, N., Ahmed, M. A., Gambhir, D., Alanazi, F. K., & Alsarra, I. A. (2014). Removal of diclofenac sodium from aqueous solution using water/Transcutol/ethylene glycol/Capryol-90 green nanoemulsions. Journal of Molecular Liquids, 199, 102–107).
Comment 2: Are all ingredients of nanoemulsion safe for the environment, including surfactant? Has it been checked?.
Response 2: In this method, capmul is chemically mixture of capric and caprylic acid as prime components. This oil was used for RIF removal which belongs to GRAS category (generally regarded as category). Transcutol has been reported as surfactant to formulate green nanoemulsion (Shakeel, F., Haq, N., Ahmed, M. A., Gambhir, D., Alanazi, F. K., & Alsarra, I. A. (2014). Removal of diclofenac sodium from aqueous solution using water/Transcutol/ethylene glycol/Capryol-90 green nanoemulsions. Journal of Molecular Liquids, 199, 102–107). DMSOT is part of nanoemulsion used at low concentration which is essential for nanoemulsion preparation. This study is preliminary step of complete water treatment. The retained solvent (DMSOT and IPA) cab be further removed using next developed approach sequentially assembled. Moreover, IPA and DMSO need to be removed from treated water using conventional method or recent novel techniques if found. The purpose of using nanoemulsion was selective removal of RIF and amplify efficiency of conventional method. DMSO and IPA can also be removed using fractional distillation method (due to difference in their boiling point). For safety concern, this project has been extended for safe concentration of excipients used to tailor nanoemulsion in suitable animal model.
Comment 3): Please specify the method of obtaining nanoemulsions. Was it PIC, EPI or other method?
Response 3: The developed nanoemulsions were prepared as method reported before (Shakeel, F., Haq, N., Ahmed, M. A., Gambhir, D., Alanazi, F. K., & Alsarra, I. A. (2014). Removal of diclofenac sodium from aqueous solution using water/Transcutol/ethylene glycol/Capryol-90 green nanoemulsions. Journal of Molecular Liquids, 199, 102–107). In this method the oil phase was slowly titrated with aqueous phase containing Smix ratio followed by delineating several phase diagrams. Several formulations were prepared at different ratios of Smix.
Comment 4: In chapter 2.2.2 authors have written that several water in oil nanoemulsion was obtained but further in the text (chapter 2.2.5) it is described that oil globules of nanoemulsion adsorbe the drug. Please explain. What was the method of nanoemulsion type determination?
Response 4: Initially, developed formulations were w/o type as you see in table 1. Maximum content of water was up to 25%. However, dispersion of nanoemulsion with aqueous phase greater that 50% starts to transform into inversion process and becomes o/w of nanoemulsion. Thus, dispersion of nanoemulsion with known volume of aqueous phase containing drug and subsequent destabilization by heating at 60 °C results in phase separation such as separate organic oil phase and aqueous phase. Drug being lipophilic get adsorbed to CMC8 which can be used for quantitative analysis using validated UPLC method. Moreover, morphological assessment of nanoemulsion using TEM corroborated o/w type of nanoemulsion (dark phase indicates oil phase) just after dispersion with aqueous phase containing RIF. I have added TEM method and result in revised manuscript.
Comment 5: In chapter 2.2.3 it is written: „Prepared GNEs were evaluated for its stability (thermodynamic), globular size, size distribution (PDI), zeta potential, refractive index (RI), and “η”. I believe that by “η” Authors meant viscosity?
Response 5: Yes. The used symbol “η” stands for viscosity. I have revised the manuscript accordingly. The changes were addressed with red text.
Comment 6: Authors have written that thixotropic behaviour was studied but I didn’t see in the manuscript any results according to those studies.
Response 6: This was typo error. The word was “rheological behaviour” used for viscosity. I have corrected for this.
Comment 7: The tables are numbered incorrect.
Response 7: I have corrected them in the revised manuscript.
Comment 8: Figure 1 should be added to the introduction section when the drug properties are described or to the Materials section.
Response 8: I have mentioned figure 1 in introduction section as per suggestion.
Comment 9: Table 1 (which should be nr 2). There is a mistake. DMSO should be instead of ethanol I believe.
Response 9: This was typo error. In trial formulation neat transcutol was not sufficient to formulate stable nanoemulsion. Therefore, DMSO containing constant concentration of transcutol (10%v/v of DMSO) was used as indicated as “DMSOT”. Therefore, I have revised the manuscript as “Several GNE (water in oil) were prepared using water, DMSOT (DMSO containing constant amount of transcutol, 10%v/v), isopropyl alcohol (IPA), and CMC8 as per reported method [20, 35]. Several batches of green nanoemulsions were prepared using oil phase titration method [35]. Notably, various trial nanoemulsions were developed using neat transcutol, IPA, and capmul. However, these were not poorly stable. Therefore, DMSO containing 10% transcutol (DMSOT) and IPA worked as suitable surfactant and co-surfactant in preparing nanoemulsions, respectively”. Moreover, title was revised to avoid confusion to readers.
Comment10: Chapter 3.2. Authors have written: „GNEs are well established carrier system and considered as thermodynamically stable system”. This information is incorrect. Nanoemulsions are kinetically stable contrary to microemulsions which are thermodynamically stable. This is due to the high concentration of surfactant in microemulsion which lowering interfacial tension to the value close “zero”. In nanoemulsion we have much lower surfactant concentration (few percent) that is why we observed destabilization processes in time (stable in time = kinetically stable). Authors should make sure if obtained by them systems are nano – or micro-emulsions (which also have nanosized particles). Especially that they have 15% of surfactant in systems which is quite high concentration. Stability studies should also cover droplet size analysis in time.
Response 11: Several authors reported nanoemulsion as thermodynamically stable system with droplet size less or near 100 nm [http://dx.doi.org/10.1016/j.molliq.2014.08.030, doi: 10.2166/wst.2014.400, http://dx.doi.org/10.1016/j.molliq.2014.12.035, http://dx.doi.org/10.1080/19443994.2014.972985, DOI: 10.1080/01496395.2014.928893]. Considering these published reports and claims, our nanoemulsions were supposed to be thermodynamically stable as they passed series of cycles of cooling and heating, and centrifugation steps. Moreover, they have globular size less than 100 nm as per these literature support. However, I have replaced the word “thermodynamically stable” with “kinetically stable”. I think this suits better theoretically as you explained. These changes were addressed with red coloured text. TEM report ensured the formation of nanoemulsion. This is a major sanctioned project and we are working several variables and removal of remained solvent in aqueous phase too. Therefore, we published data obtained from preliminary findings in this article. I will cover several variables for comprehensive stability of nanoemulson before and after dispersion with aqueous phase. I hope that portion would be published in our ensuing report.
Comment 11: By „globule size” Authors menat diameter (d) or radius of the droplets?
Response 11: Zetasizer provides diameter of droplets “d”. Therefore, globule size means diameter.
Comment 12: According to above comments figure 7 in my opinion does not sufficiently illustrate the mechanism of the process. I believe that first the drug was in water, than nanoemulsion was add and than drug adsorbed on the globules. But which globules if this is water-in-oil nanoemulsion and globules of water are dispersed in the oil continous phase…..Please clarify
Response 12: In figure 7, I have corrected by replacing the word “water-RIF precipitate” by “CMC8-RIF precipitate”. This is a proposed illustration. Yes, first the drug was in water. The w/o type of nanoemulsion was added to the aqueous solution containing RIF. Due to lipophilic nature of RIF, the drug was adsorbed to oil globules and the oil phase was destabilized by heating to high temperature as written in revised manuscript. This destabilization process leaves water phase as treated water. Therefore, I have replaced the figure 7.
Reviewer 2 Report
The manuscript “Application of Green Nanoemulsion Composed of Water/DMSO/Isopropyl Alcohol/Capmul MCM C8 for Elimination of Rifampicin from a Bulk Aqueous Solution” reports on novel multi-components green nanoemulsions based on water, IPA, DMSO, CMC8 with the aim of removing rifampicin from contaminated aqueous bulk solutions. The manuscript is well written and organized, supported by sufficient amount of data. I recommend the publication after some minor revisions and answers to some doubts/questions, reported below:
- The authors claim that the nanoemulsions are green. However, among the materials used for their preparation, DMSO, methanol and acetonitrile are not properly green solvents. The author should specify what they intend for green.
- The introduction should be enriched with more references and methods used for removing pollutants or drugs from water, in a more general context. The employment of electrostatic adsorption methods (e.g. with triboelectric devices), nanofiltration with electrospun membranes, antibiofouling coatings, or with other natural sources should be mentioned: here are some suggested references:
- Energy Technology 2017, 6(4), 10.1002/ente.201700609
- Molecules 2020, 25(16), 10.3390/molecules25163639
- Journal of Membrane Science 2013, 429:121-129, 10.1016/j.memsci.2012.11.022
- Nanomaterials and Nanotechnology 2019, 9, 10.1177/1847980419862075
- Chemical Engineering Journal 2016, 307, 10.1016/j.cej.2016.08.125
- ACS Appl. Mat. & Interf. 2021, 10.1021/acsami.1c01740
- A real photo or a TEM/SEM micrograph of the nanoemulsions would be useful to understand better the result of the preparation method.
- The conclusion should be enriched with more practical applications of the proposed methods and brief indication of future perspectives.
Author Response
Response letter Date: 15-05-2021
Reviewer 2
Comments and Suggestions for Authors
The manuscript “Application of Green Nanoemulsion Composed of Water/DMSO/Isopropyl Alcohol/Capmul MCM C8 for Elimination of Rifampicin from a Bulk Aqueous Solution” reports on novel multi-components green nanoemulsions based on water, IPA, DMSO, CMC8 with the aim of removing rifampicin from contaminated aqueous bulk solutions. The manuscript is well written and organized, supported by sufficient amount of data. I recommend the publication after some minor revisions and answers to some doubts/questions, reported below:
Comment 1: The authors claim that the nanoemulsions are green. However, among the materials used for their preparation, DMSO, methanol and acetonitrile are not properly green solvents. The author should specify what they intend for green.
Response 1: In material section, methanol and acetonitrile were used for analysis purpose. These solvents were not used in nanoemulsion. Capmul is oil used for RIF removal belongs to GRAS category (generally regarded as category). DMSOT (DMSO containing fixed concentration of transcutol) is part of nanoemulsion used at low concentration which is essential for nanoemulsion preparation. This study is preliminary step of complete water treatment. The retained solvent (DMSOT and IPA) cab be further removed using next developed approach sequentially assembled.
Comment 2: The introduction should be enriched with more references and methods used for removing pollutants or drugs from water, in a more general context. The employment of electrostatic adsorption methods (e.g. with triboelectric devices), nanofiltration with electrospun membranes, antibiofouling coatings, or with other natural sources should be mentioned: here are some suggested references:
- Energy Technology 2017, 6(4), 10.1002/ente.201700609
- Molecules 2020, 25(16), 10.3390/molecules25163639
- Journal of Membrane Science 2013, 429:121-129, 10.1016/j.memsci.2012.11.022
- Nanomaterials and Nanotechnology 2019, 9, 10.1177/1847980419862075
- Chemical Engineering Journal 2016, 307, 10.1016/j.cej.2016.08.125
- ACS Appl. Mat. & Interf. 2021, 10.1021/acsami.1c01740
Response 2: I have revised introduction section with suggested articles. Moreover, these research findings were also included in new section of “future perspectives”.
Comment 3: A real photo or a TEM/SEM micrograph of the nanoemulsions would be useful to understand better the result of the preparation method.
Response 3: I have included HR-TEM (High resolution transmission electron microscopy) images after performing experiment. NF5 was dispersed with plain water and left for 30 min. The sample was analysed for morphological assessment of nanoglobules under HR-TEM. Similarly, morphology was scanned for NF5 dispersed in the drug solution. Results showed apparently dispersed spherical nanoglobules with size below 100nm and free from sign of any instability. Dark black colour of nanoglobules are due to CMC8 as oily nature (electron beam intensity is adsorbed being lipophilic and partial drying location). These changes have been addressed in revised manuscript.
Comment 4: The conclusion should be enriched with more practical applications of the proposed methods and brief indication of future perspectives.
Response 4: The conclusion portion was revised as per suggestion. I have added an additional section as “future perspectives”.
- Kushwaha HS, Kumar A, Kumar R, Vaish R. A Water-Driven Triboelectric Generator for Electrocatalytic Wastewater Treatment. Energy Technology. 2018; 6(4): 670–676. doi:10.1002/ente.201700609.
- Encarnação T, Palito C, Pais AACC, Valente AJM, Burrows HD. Removal of Pharmaceuticals from Water by Free and Imobilised Microalgae. Molecules. 2020; 25(16): 3639. doi:10.3390/molecules25163639.
- Zaviska F, Drogui P, Grasmick A, Azais A, Héran M. Nanofiltration membrane bioreactor for removing pharmaceutical compounds. Journal of Membrane Science. 2013; 429: 121–129. doi:10.1016/j.memsci.2012.11.022.
- Mariello M, Guido F, Mastronardi VM, De Donato F, Salbini M, Brunetti V, Qualtieri A, Rizzi F, De Vittorio M. Captive-air-bubble aerophobicity measurements of antibiofouling coatings for underwater MEMS devices. Nanomaterials and Nanotechnology. 2019; 9: 184798041986207. doi:10.1177/1847980419862075.
- Bae J, Baek I, Choi H. Efficacy of piezoelectric electrospun nanofiber membrane for water treatment. Chemical Engineering Journal. 2017; 307: 670–678. doi: 10.1016/j.cej.2016.08.125.
- ACS Appl. Mat. & Interf. 2021, 10.1021/acsami.1c01740 (not available yet).
- Rodolfi, L., Chini Zittelli, G., Bassi, N., Padovani, G., Biondi, N., Bonini, G., Tredici, M.R., 2009. Microalgae for oil: strain se-lection, induction of lipid synthesis and outdoor mass cultivation in a low-cost photobioreactor. Biotechnol. Bioeng. 102, 100-112.

Reviewer 3 Report
The paper presents results on the preparation and characterization [(water/dimethyl sulfoxide/isopropyl alcohol/capmul MCM C8 (CMC8)] of green nanoemulsions and their use for the removal of Rifampicin from aqueous solution.
Authors reports a quite extensive set of results concerning the characterization of the GNE and correlate their properties with removal efficiency.
There are some important issues that must be addressed and evidences must be provided regarding the liquid-liquid extraction system used:
- How stable is the GNE in water?
- What is the composition of the organic phase that you recover in the supernatant?
- Can you be sure that the dimethyl sulfoxide or the isopropyl alcohol is not released (extracted) to the aqueous media during the extraction procedure?
These are central questions that may justify the reduced number of papers on the use of GNE for contaminants removal.
Besides these questions, there are other aspects of the paper that should be improved.
- The zeta potential results (that are referred in the abstract) are not displayed in the paper.
- “Some-times, few drug or pharmaceutical products have not been anticipated to cause serious threats (disease) to human health but it may be a causative factor for harmful effect to the environment due to scarcity of eco-toxicological data [1]. “ – it does not seem plausible that the scarcity of eco-toxicological data is responsible for the harmful effects of substances for the environment. The sentence must be reformulated in order to clarify its meaning.
- In the methods section I couln’t understand what is “(1 g as “m”)” for the amount of GNE used for the extraction of RIF from 10 ml of aqueous solution.
- You should provide evidences that support the proposed extraction mechanism, namely concerning the precipitation of RIF is the GNE.
Suggestion: The characterization of the supernatant after the extraction could provide relevant information regarding the issues addressed in i), ii), iii) and 4.
Author Response
Response letter Date: 15-05-2021
Reviewer 3
Comments and Suggestions for Authors
The paper presents results on the preparation and characterization [(water/dimethyl sulfoxide/isopropyl alcohol/capmul MCM C8 (CMC8)] of green nanoemulsions and their use for the removal of Rifampicin from aqueous solution.
Authors reports a quite extensive set of results concerning the characterization of the GNE and correlate their properties with removal efficiency.
There are some important issues that must be addressed and evidences must be provided regarding the liquid-liquid extraction system used:
Comment 1: How stable is the GNE in water?
Response 1: Nanoemulsions are thermodynamically stable systems as compared to dispersion system and emulsions. Therefore, the prime objective was to remove unstable or metastable formed system by thermodynamic stability test (centrifugation, and cycles of heating and cooling steps). We also investigated self-emulsification efficiency test using aqueous system as dispersion medium (data not included in this article). Nanoemulsions exhibiting no sign of precipitation or phase separation were categorized a grade “A” and considered as “passed”. Moreover, after passing through cycles of heating and cooling phase, nanoemulsions were stable and resumed transparent isotropic clear solution. This was stable in water phase due to self-emulsification efficiency achieved through surfactant and co-surfactant proper ratio “Smix”. Therefore, phase diagram study was conducted to select stable composition of nanoemulsion even after dispersing in aqueous phase. Few compositions were not stable and they were not considered suitable in this study. In this revision, DMSO containing fixed concentration of transcutol (10%) was expressed as DMSOT. Therefore, the missing transcutol was revised in the revision.
Comment 2: What is the composition of the organic phase that you recover in the supernatant?
Response 2: The destabilized nanoemulsion (by heating to high temperature for 2 h) by centrifugation gave clear aqueous supernatant and oily phase at bottom containing RIF. RIF was analysed. I have corrected the sentence in the revised manuscript. The organic phase mainly contains CMC8 in the range of 25-37%.
Comment 3: Can you be sure that the dimethyl sulfoxide or the isopropyl alcohol is not released (extracted) to the aqueous media during the extraction procedure?
Response 3: The proposed study is related to selective removal of toxic rifampicin using minimum content of DMSO (containing transcutol expressed as DMSOT) or IPA in aqueous phase. Moreover, this is preliminary treatment of contaminated water which needs to be treated in second stage. In further process, volatile IPA and soluble DMSOT were removed fractional distillation technique.
These are central questions that may justify the reduced number of papers on the use of GNE for contaminants removal.
Besides these questions, there are other aspects of the paper that should be improved.
Comment 4: The zeta potential results (that are referred in the abstract) are not displayed in the paper.
Response 4: The values of zeta potential have been added in table 1.
Comment 5: “Sometimes, few drug or pharmaceutical products have not been anticipated to cause serious threats (disease) to human health but it may be a causative factor for harmful effect to the environment due to scarcity of eco-toxicological data [1]. “ – it does not seem plausible that the scarcity of eco-toxicological data is responsible for the harmful effects of substances for the environment. The sentence must be reformulated in order to clarify its meaning.
Response 5: Yes. I have revised the sentence as per suggestion.
Comment 6: In the methods section I couln’t understand what is “(1 g as “m”)” for the amount of GNE used for the extraction of RIF from 10 ml of aqueous solution.
Response 6: I have corrected the section where “m” represented the weighed amount of nanoemulsion taken (m = 1 g) for dispersion in 10 ml of RIF solution. The equation 1 required the value of “m” for calculation.
Comment 7: You should provide evidences that support the proposed extraction mechanism, namely concerning the precipitation of RIF is the GNE.
Response 7: I have illustrated proposed mechanism of removal of RIF from contaminated water. In brief, RIF is lipophilic drug and insoluble in aqueous medium above 1-2 mg/ml concentration at room temperature. After dispersion with bulk aqueous solution, RIF was probably adsorbed to the oil phase of nanoemulsion due to preferential adsorption and lipophilic –lipophilic molecular interaction. In this method, nanoemulsion was initially dispersed with aqueous solution of RIF and subjected to freeze thaw cycles (-21 °C for 5 h) followed by keeping at 60 °C for 3 h (hot air oven) to destabilize nanoemulsion. Oil phase loaded with RIF was separated out by centrifugation and quantified using UPLC method. The separated aqueous phase was also analysed and revealed no peak at their peak retention time suggesting removal of RIF form aqueous system.
Comment 8: Suggestion: The characterization of the supernatant after the extraction could provide relevant information regarding the issues addressed in i), ii), iii) and 4.
Response 8: I have added a new section 3.5 in result and discussion. The presence of DMSO and transcutol were either extremely low or zero. Therefore, UV and IR based estimations were insignificant. To confirm probable chance of DMSO extraction in the treated water, SEM-EDX and ICP-OE methods were used to detect the presence of “S” sulphur as an element of DMSO. I have included these findings which suggested absence of DMSO extraction into the treated water. I added these findings as Chemically, DMSO is dimethyl sulfoxide containing sulphur as an element. ICP-OE was used to detect presence of elements in the treated water. It was expected that the presence of sulphur in treated water may confirm extraction of DMSO from nanoemulsion. C, H, and O are common elements present in water, transcutol, and DMSO. Therefore, these elements were kept out of detection due to interference in results. Similarly, SEM-EDX method was used to detect elements excluding the common elements. The results are presented in figure 9. Figure 9A is the result of SEM-EDX which exhibited several peaks of intensity of elements present in the treated water. The most intense peak was observed for Ca, Na, and K. This qualitative assessment confirmed that these alkali and alkali like metals are due to hardness of water (dissolved salt). However, Fe, Ti, and Cu were extremely low. Cu may be due to copper grid used during analysis. Ti and Fe were insignificant. Notably, the treated was free from any sulphur element suggesting absence of DMSO extraction or the content might be below detection. These findings were further supported with quantitative assessment using ICP-OE.
The result of ICP-OE has been portrayed in figure 9B where Ca (0.089 ppm), K (0.004ppm), Mg (0.042) and Na (0.048) were considerably high in the treated water. However, Fe, Ti, Zn, Mo, and Cu were below detection limit (BDL). The absence sulphur in figure 9B confirmed absence of DMSO extraction below detection limit. The treated water was also scanned using UV Vis spectrophotometer and IR spectroscopy to identify absorbance wavelength and finger printing of functional groups of DMSO, transcutol, and other component, respectively (data not included here). These approach could not detected due to the content either below BDL or absence of these. Conclusively, the separated organic phase included CMC8, and Smix leaving water phase behind”.
Round 2
Reviewer 1 Report
Reviewer 1
Comment 1): Why DMSO was chosen as a surfactant? Is there any information in the literature about DMSO- stabilized nanoemulsions? Does it have proven and sufficient surface active properties? If yes, please cite appropriate literature.
Response 1: Thank you for pointing out missing content. In this study, DMSO contains a fixed concentration of transcutol as surfactant (10%v/v, and expressed as DMSOT). Therefore, experimental section was revised in the revised manuscript. Moreover, the title was revised to avoid this confusion. During exhaustive phase diagram study, we come to know, neat transcutol was capable to give limited zone of nanemulsion (data not included in this study). Faiyaz et al. reported successful formation of green nanoemulsion to remove diclofenac from aqueous solution (Shakeel, F., Haq, N., Ahmed, M. A., Gambhir, D., Alanazi, F. K., & Alsarra, I. A. (2014). Removal of diclofenac sodium from aqueous solution using water/Transcutol/ethylene glycol/Capryol-90 green nanoemulsions. Journal of Molecular Liquids, 199, 102–107).
OK
Comment 2: Are all ingredients of nanoemulsion safe for the environment, including surfactant? Has it been checked?.
Response 2: In this method, capmul is chemically mixture of capric and caprylic acid as prime components. This oil was used for RIF removal which belongs to GRAS category (generally regarded as category). Transcutol has been reported as surfactant to formulate green nanoemulsion (Shakeel, F., Haq, N., Ahmed, M. A., Gambhir, D., Alanazi, F. K., & Alsarra, I. A. (2014). Removal of diclofenac sodium from aqueous solution using water/Transcutol/ethylene glycol/Capryol-90 green nanoemulsions. Journal of Molecular Liquids, 199, 102–107). DMSOT is part of nanoemulsion used at low concentration which is essential for nanoemulsion preparation. This study is preliminary step of complete water treatment. The retained solvent (DMSOT and IPA) cab be further removed using next developed approach sequentially assembled. Moreover, IPA and DMSO need to be removed from treated water using conventional method or recent novel techniques if found. The purpose of using nanoemulsion was selective removal of RIF and amplify efficiency of conventional method. DMSO and IPA can also be removed using fractional distillation method (due to difference in their boiling point). For safety concern, this project has been extended for safe concentration of excipients used to tailor nanoemulsion in suitable animal model.
OK
Comment 3): Please specify the method of obtaining nanoemulsions. Was it PIC, EPI or other method?
Response 3: The developed nanoemulsions were prepared as method reported before (Shakeel, F., Haq, N., Ahmed, M. A., Gambhir, D., Alanazi, F. K., & Alsarra, I. A. (2014). Removal of diclofenac sodium from aqueous solution using water/Transcutol/ethylene glycol/Capryol-90 green nanoemulsions. Journal of Molecular Liquids, 199, 102–107). In this method the oil phase was slowly titrated with aqueous phase containing Smix ratio followed by delineating several phase diagrams. Several formulations were prepared at different ratios of Smix.
- In table 1 there is still mistake. There should be DMSOT instead of ethanol.
Comment 4: In chapter 2.2.2 authors have written that several water in oil nanoemulsion was obtained but further in the text (chapter 2.2.5) it is described that oil globules of nanoemulsion adsorbe the drug. Please explain. What was the method of nanoemulsion type determination?
Response 4: Initially, developed formulations were w/o type as you see in table 1. Maximum content of water was up to 25%. However, dispersion of nanoemulsion with aqueous phase greater that 50% starts to transform into inversion process and becomes o/w of nanoemulsion. Thus, dispersion of nanoemulsion with known volume of aqueous phase containing drug and subsequent destabilization by heating at 60 °C results in phase separation such as separate organic oil phase and aqueous phase. Drug being lipophilic get adsorbed to CMC8 which can be used for quantitative analysis using validated UPLC method. Moreover, morphological assessment of nanoemulsion using TEM corroborated o/w type of nanoemulsion (dark phase indicates oil phase) just after dispersion with aqueous phase containing RIF. I have added TEM method and result in revised manuscript.
It is still unclear. You have obtained W/O nanoemulsion which containg lipophilic drug in external phase. Right? Why did you dispersed in water and drug dispersion W/O nanoemulsion and made TEM analysis? In the text you have wrote: „NF5 could not exhibit anysign of Oswald ripening”. But those studies are only after 30 min so you can not observed phase separation at so short time.I believe that those studies was aimed to how the GNE will behave in aquatic ecosystem? But in that case you obtained new system and you can not write that „The morphological assessment of NF5 was carried out using HR-TEM after dispersion”. This is new system after phase inversion. Data form figure 3 are not correlated with table 3. On the figure 3 picture you have O/W nanoemulsion (after dispersion O/W nanoemulsion in water) and in the table 3 W/O nanoemulsion’s size. It should be clarify.
Comment 5: In chapter 2.2.3 it is written: „Prepared GNEs were evaluated for its stability (thermodynamic), globular size, size distribution (PDI), zeta potential, refractive index (RI), and “η”. I believe that by “η” Authors meant viscosity?
Response 5: Yes. The used symbol “η” stands for viscosity. I have revised the manuscript accordingly. The changes were addressed with red text.
OK
Comment 6: Authors have written that thixotropic behaviour was studied but I didn’t see in the manuscript any results according to those studies.
Response 6: This was typo error. The word was “rheological behaviour” used for viscosity. I have corrected for this.
OK
Comment 7: The tables are numbered incorrect.
Response 7: I have corrected them in the revised manuscript.
OK
Comment 8: Figure 1 should be added to the introduction section when the drug properties are described or to the Materials section.
Response 8: I have mentioned figure 1 in introduction section as per suggestion.
I stand by my opinion that the drawing should be in te introduction.
Comment 9: Table 1 (which should be nr 2). There is a mistake. DMSO should be instead of ethanol I believe.
Response 9: This was typo error. In trial formulation neat transcutol was not sufficient to formulate stable nanoemulsion. Therefore, DMSO containing constant concentration of transcutol (10%v/v of DMSO) was used as indicated as “DMSOT”. Therefore, I have revised the manuscript as “Several GNE (water in oil) were prepared using water, DMSOT (DMSO containing constant amount of transcutol, 10%v/v), isopropyl alcohol (IPA), and CMC8 as per reported method [20, 35]. Several batches of green nanoemulsions were prepared using oil phase titration method [35]. Notably, various trial nanoemulsions were developed using neat transcutol, IPA, and capmul. However, these were not poorly stable. Therefore, DMSO containing 10% transcutol (DMSOT) and IPA worked as suitable surfactant and co-surfactant in preparing nanoemulsions, respectively”. Moreover, title was revised to avoid confusion to readers.
There is still ethanol in table 1.
Comment10: Chapter 3.2. Authors have written: „GNEs are well established carrier system and considered as thermodynamically stable system”. This information is incorrect. Nanoemulsions are kinetically stable contrary to microemulsions which are thermodynamically stable. This is due to the high concentration of surfactant in microemulsion which lowering interfacial tension to the value close “zero”. In nanoemulsion we have much lower surfactant concentration (few percent) that is why we observed destabilization processes in time (stable in time = kinetically stable). Authors should make sure if obtained by them systems are nano – or micro-emulsions (which also have nanosized particles). Especially that they have 15% of surfactant in systems which is quite high concentration. Stability studies should also cover droplet size analysis in time.
Response 11: Several authors reported nanoemulsion as thermodynamically stable system with droplet size less or near 100 nm [http://dx.doi.org/10.1016/j.molliq.2014.08.030, doi: 10.2166/wst.2014.400, http://dx.doi.org/10.1016/j.molliq.2014.12.035, http://dx.doi.org/10.1080/19443994.2014.972985, DOI: 10.1080/01496395.2014.928893]. Considering these published reports and claims, our nanoemulsions were supposed to be thermodynamically stable as they passed series of cycles of cooling and heating, and centrifugation steps. Moreover, they have globular size less than 100 nm as per these literature support. However, I have replaced the word “thermodynamically stable” with “kinetically stable”. I think this suits better theoretically as you explained. These changes were addressed with red coloured text. TEM report ensured the formation of nanoemulsion. This is a major sanctioned project and we are working several variables and removal of remained solvent in aqueous phase too. Therefore, we published data obtained from preliminary findings in this article. I will cover several variables for comprehensive stability of nanoemulson before and after dispersion with aqueous phase. I hope that portion would be published in our ensuing report.
TEM report concerning O/W type nanoemulsion not W/O nanoemulsion as I described before. I am still not sure that you have obtained nano- or microemulsions becasue O/W nanoemulsions can be obtained by diluted W/O microemulsion (one of the method).You should put it to the fridge and observed they appereance. In case of transparent nanoemulsion there will be no changes. In case of transparent microemulsion they become white in the fridge and than in the room temperaturÄ™ they become transparent again. But most of all I stand by my opinion that stability studies should also cover droplet size analysis in time.
Comment 11: By „globule size” Authors menat diameter (d) or radius of the droplets?
Response 11: Zetasizer provides diameter of droplets “d”. Therefore, globule size means diameter.
Some of Zetasizers provides diameter and some radius that is why it should be clarofy by authors.
Comment 12: According to above comments figure 7 in my opinion does not sufficiently illustrate the mechanism of the process. I believe that first the drug was in water, than nanoemulsion was add and than drug adsorbed on the globules. But which globules if this is water-in-oil nanoemulsion and globules of water are dispersed in the oil continous phase…..Please clarify
Response 12: In figure 7, I have corrected by replacing the word “water-RIF precipitate” by “CMC8-RIF precipitate”. This is a proposed illustration. Yes, first the drug was in water. The w/o type of nanoemulsion was added to the aqueous solution containing RIF. Due to lipophilic nature of RIF, the drug was adsorbed to oil globules and the oil phase was destabilized by heating to high temperature as written in revised manuscript. This destabilization process leaves water phase as treated water. Therefore, I have replaced the figure 7.
First of all you should add on the figure heat source and temperature (600C) and that nanoemulsion z W/O was added to the aqueous system with RIF particles (left side of the figure). W/O nanoemulsion was added to the aqueos phase with RIF and than RIF adsorbed on the oil globulus (particles of RIF should be also ilustrated in water on the left part of figure. Than as you mentioned O/W nanoemulsion (or not?) was obtained (TEM was obtained for that system) which is not clear on the figure. And than you have destabilized the O/W system and separate oil (with RIF) phase from water phase. It is still not clear in my opinion.
Author Response
Response letter Date: 25-05-2021
Reviewer 1
Comment 1): Why DMSO was chosen as a surfactant? Is there any information in the literature about DMSO- stabilized nanoemulsions? Does it have proven and sufficient surface active properties? If yes, please cite appropriate literature.
Response 1: Thank you for pointing out missing content. In this study, DMSO contains a fixed concentration of transcutol as surfactant (10%v/v, and expressed as DMSOT). Therefore, experimental section was revised in the revised manuscript. Moreover, the title was revised to avoid this confusion. During exhaustive phase diagram study, we come to know, neat transcutol was capable to give limited zone of nanemulsion (data not included in this study). Faiyaz et al. reported successful formation of green nanoemulsion to remove diclofenac from aqueous solution (Shakeel, F., Haq, N., Ahmed, M. A., Gambhir, D., Alanazi, F. K., & Alsarra, I. A. (2014). Removal of diclofenac sodium from aqueous solution using water/Transcutol/ethylene glycol/Capryol-90 green nanoemulsions. Journal of Molecular Liquids, 199, 102–107).
OK
Comment 2: Are all ingredients of nanoemulsion safe for the environment, including surfactant? Has it been checked?.
Response 2: In this method, capmul is chemically mixture of capric and caprylic acid as prime components. This oil was used for RIF removal which belongs to GRAS category (generally regarded as category). Transcutol has been reported as surfactant to formulate green nanoemulsion (Shakeel, F., Haq, N., Ahmed, M. A., Gambhir, D., Alanazi, F. K., & Alsarra, I. A. (2014). Removal of diclofenac sodium from aqueous solution using water/Transcutol/ethylene glycol/Capryol-90 green nanoemulsions. Journal of Molecular Liquids, 199, 102–107). DMSOT is part of nanoemulsion used at low concentration which is essential for nanoemulsion preparation. This study is preliminary step of complete water treatment. The retained solvent (DMSOT and IPA) cab be further removed using next developed approach sequentially assembled. Moreover, IPA and DMSO need to be removed from treated water using conventional method or recent novel techniques if found. The purpose of using nanoemulsion was selective removal of RIF and amplify efficiency of conventional method. DMSO and IPA can also be removed using fractional distillation method (due to difference in their boiling point). For safety concern, this project has been extended for safe concentration of excipients used to tailor nanoemulsion in suitable animal model.
OK
Comment 3): Please specify the method of obtaining nanoemulsions. Was it PIC, EPI or other method?
Response 3: The developed nanoemulsions were prepared as method reported before (Shakeel, F., Haq, N., Ahmed, M. A., Gambhir, D., Alanazi, F. K., & Alsarra, I. A. (2014). Removal of diclofenac sodium from aqueous solution using water/Transcutol/ethylene glycol/Capryol-90 green nanoemulsions. Journal of Molecular Liquids, 199, 102–107). In this method the oil phase was slowly titrated with aqueous phase containing Smix ratio followed by delineating several phase diagrams. Several formulations were prepared at different ratios of Smix.
- In table 1 there is still mistake. There should be DMSOT instead of ethanol.
Response 3: I have corrected table 1.
Comment 4: In chapter 2.2.2 authors have written that several water in oil nanoemulsion was obtained but further in the text (chapter 2.2.5) it is described that oil globules of nanoemulsion adsorbe the drug. Please explain. What was the method of nanoemulsion type determination?
Response 4: Initially, developed formulations were w/o type as you see in table 1. Maximum content of water was up to 25%. However, dispersion of nanoemulsion with aqueous phase greater that 50% starts to transform into inversion process and becomes o/w of nanoemulsion. Thus, dispersion of nanoemulsion with known volume of aqueous phase containing drug and subsequent destabilization by heating at 60 °C results in phase separation such as separate organic oil phase and aqueous phase. Drug being lipophilic get adsorbed to CMC8 which can be used for quantitative analysis using validated UPLC method. Moreover, morphological assessment of nanoemulsion using TEM corroborated o/w type of nanoemulsion (dark phase indicates oil phase) just after dispersion with aqueous phase containing RIF. I have added TEM method and result in revised manuscript.
It is still unclear. You have obtained W/O nanoemulsion which containg lipophilic drug in external phase. Right? Why did you dispersed in water and drug dispersion W/O nanoemulsion and made TEM analysis? In the text you have wrote: „NF5 could not exhibit anysign of Oswald ripening”. But those studies are only after 30 min so you can not observed phase separation at so short time. I believe that those studies was aimed to how the GNE will behave in aquatic ecosystem? But in that case you obtained new system and you can not write that „The morphological assessment of NF5 was carried out using HR-TEM after dispersion”. This is new system after phase inversion. Data form figure 3 are not correlated with table 3. On the figure 3 picture you have O/W nanoemulsion (after dispersion O/W nanoemulsion in water) and in the table 3 W/O nanoemulsion’s size. It should be clarify.
Response 4: Firstly, Dear sir. With due respect I would like to clear one thing. There is little bit confusion in you regarding presented nanoemulsion in the manuscript and the purpose of water cleaning contaminated with rifampicin. Let me clear this point to you and your raised comments point to point. Nanoemulsions presented in table 1 are w/o type without loading rifampicin. This is what raised first question. There is no drug incorporated in external phase (oil). This is the purpose of preparing w/o type of nanoemulsion which may be used to decontaminate an aqueous system containing rifampicin at trace level. Therefore, we had prepared a stock aqueous solution of rifampicin in which w/o nanoemulsion was dispersed and there is phase inversion after dispersion due to bulk volume of aqueous phase relative to oil phase. Please note, at this time drug get diffused in oil phase (due to possible lipophilic lipophilic interaction and lipophilic nature of RIF). It means nanoemulsion after dispersion (o/w) contains rifampicin in lipid phase which may remain stable at room temperature. However, it may exhibit Oswald ripening if kept for long time (I said in this context). Just to avoid confusion to reader, I removed the sentence about Oswald ripening in the revised manuscript. But, in this study, this nanoemulsion was forcibly subjected to phase separation by freezing followed by heating at high temperature as I mentioned in revised manuscript and I also cited a new reference [37, (Shakeel, F., Haq, N., Alanazi, F. K., & Alsarra, I. A. (2014). Removal of methyl orange from aqueous solution by EA/Triton-X100/EG/water green nanoemulsions. Desalination and Water Treatment, 57(2), 747–753. doi:10.1080/19443994.2014.972985) for this phase separation method)].
Secondly, Yes. Thankful for pointing out an important mistake “The morphological assessment of NF5 was carried out using HR-TEM after dispersion”. This seems to create confusion to reader. After dispersion, NF5 is in o/w type (after phase inversion) not w/o. Therefore, I should revise the sentence as “The morphological assessment of dispersed NF5 was carried out using HR-TEM without destabilization”. I have revised the manuscript in text body and figure caption (TEM) accordingly. These changes were highlighted. I presented new system as “dispersed NF5”. I still kept NF5 because of NF5 dispersion into the drug solution and plain water. I hope there may not be any confusion to reader.
Thirdly, Dear sir, figure 3 cannot be correlated with table 3 in term of globular size, and PDI values. The nanoemulsion (NF5) is w/o type of nanoemulsion in table 3 whereas the dispersed NF5 (either in plain water or the aqueous drug solution) is o/w type of nanoemulsion. The TEM images corroborated successful conversion of NF5 from w/o type to o/w type. Moreover, the globular size observed in TEM was comparable to ~ 38.0 nm and spherical in shape. The globules were well dispersed in water system (as observed in figure 3).
Comment 5: In chapter 2.2.3 it is written: „Prepared GNEs were evaluated for its stability (thermodynamic), globular size, size distribution (PDI), zeta potential, refractive index (RI), and “η”. I believe that by “η” Authors meant viscosity?
Response 5: Yes. The used symbol “η” stands for viscosity. I have revised the manuscript accordingly. The changes were addressed with red text.
OK
Comment 6: Authors have written that thixotropic behaviour was studied but I didn’t see in the manuscript any results according to those studies.
Response 6: This was typo error. The word was “rheological behaviour” used for viscosity. I have corrected for this.
OK
Comment 7: The tables are numbered incorrect.
Response 7: I have corrected them in the revised manuscript.
OK
Comment 8: Figure 1 should be added to the introduction section when the drug properties are described or to the Materials section.
Response 8: I have mentioned figure 1 in introduction section as per suggestion.
I stand by my opinion that the drawing should be in te introduction.
Response: I have mentioned figure 1 in material section.
Comment 9: Table 1 (which should be nr 2). There is a mistake. DMSO should be instead of ethanol I believe.
Response 9: This was typo error. In trial formulation neat transcutol was not sufficient to formulate stable nanoemulsion. Therefore, DMSO containing constant concentration of transcutol (10%v/v of DMSO) was used as indicated as “DMSOT”. Therefore, I have revised the manuscript as “Several GNE (water in oil) were prepared using water, DMSOT (DMSO containing constant amount of transcutol, 10%v/v), isopropyl alcohol (IPA), and CMC8 as per reported method [20, 35]. Several batches of green nanoemulsions were prepared using oil phase titration method [35]. Notably, various trial nanoemulsions were developed using neat transcutol, IPA, and capmul. However, these were not poorly stable. Therefore, DMSO containing 10% transcutol (DMSOT) and IPA worked as suitable surfactant and co-surfactant in preparing nanoemulsions, respectively”. Moreover, title was revised to avoid confusion to readers.
There is still ethanol in table 1.
Response: I have corrected table 1.
Comment10: Chapter 3.2. Authors have written: „GNEs are well established carrier system and considered as thermodynamically stable system”. This information is incorrect. Nanoemulsions are kinetically stable contrary to microemulsions which are thermodynamically stable. This is due to the high concentration of surfactant in microemulsion which lowering interfacial tension to the value close “zero”. In nanoemulsion we have much lower surfactant concentration (few percent) that is why we observed destabilization processes in time (stable in time = kinetically stable). Authors should make sure if obtained by them systems are nano – or micro-emulsions (which also have nanosized particles). Especially that they have 15% of surfactant in systems which is quite high concentration. Stability studies should also cover droplet size analysis in time.
Response 11: Several authors reported nanoemulsion as thermodynamically stable system with droplet size less or near 100 nm [http://dx.doi.org/10.1016/j.molliq.2014.08.030, doi: 10.2166/wst.2014.400, http://dx.doi.org/10.1016/j.molliq.2014.12.035, http://dx.doi.org/10.1080/19443994.2014.972985, DOI: 10.1080/01496395.2014.928893]. Considering these published reports and claims, our nanoemulsions were supposed to be thermodynamically stable as they passed series of cycles of cooling and heating, and centrifugation steps. Moreover, they have globular size less than 100 nm as per these literature support. However, I have replaced the word “thermodynamically stable” with “kinetically stable”. I think this suits better theoretically as you explained. These changes were addressed with red coloured text. TEM report ensured the formation of nanoemulsion. This is a major sanctioned project and we are working several variables and removal of remained solvent in aqueous phase too. Therefore, we published data obtained from preliminary findings in this article. I will cover several variables for comprehensive stability of nanoemulson before and after dispersion with aqueous phase. I hope that portion would be published in our ensuing report.
TEM report concerning O/W type nanoemulsion not W/O nanoemulsion as I described before. I am still not sure that you have obtained nano- or microemulsions becasue O/W nanoemulsions can be obtained by diluted W/O microemulsion (one of the method).You should put it to the fridge and observed they appereance. In case of transparent nanoemulsion there will be no changes. In case of transparent microemulsion they become white in the fridge and than in the room temperaturÄ™ they become transparent again. But most of all I stand by my opinion that stability studies should also cover droplet size analysis in time.
Response: Dear sir, I described again this response letter about first w/o nanoemulsion (table 1) and its (NF5) conversion into o/w type of nanoemulsion. I also revised this manuscript to avoid confusion. These changes have been addressed in revised R2 file. I had studied freeze thaw cycles at two extreme temperatures (-21 and 40 °C) and resuming transparency at 25 °C as shown in table 2. There were no signs of physical instability like phase separation.
Stability study of nanoemulsion is under process due to long term study. Therefore, we have not included in this manuscript.
Comment 11: By „globule size” Authors menat diameter (d) or radius of the droplets?
Response 11: Zetasizer provides diameter of droplets “d”. Therefore, globule size means diameter.
Some of Zetasizers provides diameter and some radius that is why it should be clarofy by authors.
Response: I have mentioned in revised manuscript. It was radius in results (r.nm).
Comment 12: According to above comments figure 7 in my opinion does not sufficiently illustrate the mechanism of the process. I believe that first the drug was in water, than nanoemulsion was add and than drug adsorbed on the globules. But which globules if this is water-in-oil nanoemulsion and globules of water are dispersed in the oil continous phase…..Please clarify
Response 12: In figure 7, I have corrected by replacing the word “water-RIF precipitate” by “CMC8-RIF precipitate”. This is a proposed illustration. Yes, first the drug was in water. The w/o type of nanoemulsion was added to the aqueous solution containing RIF. Due to lipophilic nature of RIF, the drug was adsorbed to oil globules and the oil phase was destabilized by heating to high temperature as written in revised manuscript. This destabilization process leaves water phase as treated water. Therefore, I have replaced the figure 7.
First of all you should add on the figure heat source and temperature (600C) and that nanoemulsion z W/O was added to the aqueous system with RIF particles (left side of the figure). W/O nanoemulsion was added to the aqueos phase with RIF and than RIF adsorbed on the oil globulus (particles of RIF should be also ilustrated in water on the left part of figure. Than as you mentioned O/W nanoemulsion (or not?) was obtained (TEM was obtained for that system) which is not clear on the figure. And than you have destabilized the O/W system and separate oil (with RIF) phase from water phase. It is still not clear in my opinion.
Response: I have replaced the figure 8 with more simplified schematic illustration.
Response letter Date: 23-05-2021
Reviewer 3
Comments and Suggestions for Authors
The modifications were adequate and clarified the questions raised during the paper analysis.

Reviewer 3 Report
The modifications were adequate and clarified the questions raised during the paper analysis.
Author Response
Response letter Date: 25-05-2021
Reviewer 3
Comments and Suggestions for Authors
The modifications were adequate and clarified the questions raised during the paper analysis.
Response: Thanking you for positive comments and considering responses. I have also revised the manuscript for moderate language correction. These changes have been highlighted in the revised manuscript text body.
